# Effects of Ultrasonic Surface Rolling on the Localized Corrosion Behavior of 7B50-T7751 Aluminum Alloy

**DOI:** 10.3390/ma13030738

**Published:** 2020-02-06

**Authors:** Xingchen Xu, Daoxin Liu, Xiaohua Zhang, Chengsong Liu, Dan Liu, Amin Ma

**Affiliations:** Corrosion and Protection Research Laboratory, Northwestern Polytechnical University, Xi’an 710072, China; xaxxc@163.com (X.X.); yhzhangxh@163.com (X.Z.); liu3307778@126.com (C.L.); liudansmile@mail.nwpu.edu.cn (D.L.); mam0121@126.com (A.M.)

**Keywords:** aluminum alloy, pitting, compressive residual stress, nanocrystalline, ultrasonic surface rolling process

## Abstract

The effects of an ultrasonic surface rolling process (USRP) on the localized corrosion behavior of 7B50-T7751 aluminum alloy in a sodium chloride + hydrogen peroxide solution were investigated through microstructural observation, immersion testing, and electrochemical measurements. The results revealed that this alloy is prone to pitting. However, the localized corrosion resistance can be significantly improved via both one-pass USRP and 12-pass USRP treatment. Furthermore, in the test solution, the thickness and the acceptor density of the passivation film were affected by the USRP treatment. The improved corrosion resistance of one-pass USRP-treated samples resulted mainly from the introduced compressive residual stress. However, this stress played a secondary role in the considerable enhancement observed for the corrosion resistance of the 12-pass USRP-treated samples. This enhancement is attributed primarily to the nanocrystalline surface and homogeneous surface microstructure induced by the multiple-pass USRP treatment.

## 1. Introduction

Due to their low cost, high corrosion resistance, and high specific strength, 7000 series ultra-high-strength aluminum (Al) alloys have been extensively utilized in the aerospace, automotive, and construction industries [1,2,3]. The corrosion resistance of aluminum alloys originates from the stable passivation film naturally formed on the surface [4]. However, corrosive media with high concentrations of chloride ions lead to rupture of the passive film and, consequently, both uniform and localized corrosion [5]. Compared with uniform corrosion, localized corrosion (for example, pitting corrosion, intergranular corrosion (IGC), and crevice corrosion) is more common and detrimental to aluminum alloy components [6]. Therefore, the corrosion prevention is essential and is ordinarily achieved via diverse types of coatings [3,7,8,9,10] or under conditions where mechanical and microstructural improvements are required, with different surface treatment technologies [11,12,13].

Surface severe plastic deformation (SSPD) techniques include ultrasonic shot peening [14], surface mechanical rolling treatment [15], and surface mechanical attrition treatment [16,17]. Through these widely investigated techniques, fatigue life, as well as wear and corrosion resistance of a material, can be improved without a significant loss of ductility. Recently, the ultrasonic surface rolling process (USRP), an SSPD technique, has attracted considerable attention, owing to its simplicity and ability to develop a homogeneous surface via precise control of the process parameters [18]. Furthermore, a gradient microstructure that changes gradually from the surface to the interior, without a sharp interface between the deformed layer and the substrate, can be obtained through multiple-pass USRP treatment [19,20]. The deformation mechanism and properties of some USRP-treated metallic materials have been extensively investigated. For instance, in the case of USRP-treated 17-4PH stainless steel, initially formed elongated ultrafine grains were subsequently refined via dislocation glide [21]. Zhang et al. [22] reported that a deformation layer, with a certain thickness and composed of nanoscale grains, was formed on the surface of USRP-treated 17-4PH stainless steel. The corrosion resistance of the stainless steel increased notably after the USRP treatment. Ye et al. [23] stated that the increase in the corrosion resistance of AZ31B magnesium alloy resulted mainly from the surface roughness reduction, severe plastic deformation, and surface grain refinement induced by USRP.

Our preliminary studies [24,25] revealed that the surface of the 7B50 aluminum alloy sample is contaminant-free after the USRP treatment. The compressive residual stress plays a major role in increasing the corrosion fatigue life of the USRP-treated samples (including samples with and without a nanocrystalline structure). However, a systematic study focused on the corrosion behavior of these samples is lacking. Therefore, in this work, 7B50-T7751 aluminum alloy samples with and without a nanocrystalline structure were obtained by means of a 12-pass and a one-pass USRP treatment, respectively. Subsequently, the corrosion behavior of these samples in sodium chloride (NaCl) + hydrogen peroxide (H_2_O_2_) solution was evaluated via immersion testing, potentiodynamic polarization, electrochemical impedance spectroscopy (EIS), and Mott–Schottky analysis. The effect of the compressive residual stress on the corrosion resistance was determined via the residual stress relaxation method. The present investigation was aimed at elucidating the application prospect of USRP in raising the localized corrosion resistance of aeronautical aluminum alloy structure (e.g., fuselage frame, skin, and stringer of the wing).

## 2. Materials and Methods

### 2.1. Materials and Sample Preparation

In this study, the commercial 7B50-T7751 aluminum alloy (Southwest Aluminum Group Co., Ltd., Chongqing, China) with a retrogression and re-aging treatment was employed. This alloy was received as a 40-mm-thick rolled plate (see Table 1 for the corresponding chemical composition). The mean grain size ranges were 500–2000 μm, 50–120 μm and 5–25 μm, in the longitudinal, transverse, and thickness directions, respectively. In preparation for USRP, cuboid blanks were cut along the rolling direction of the plate via a line cutting machine. These blanks were then processed into rod specimens (length: 65 mm, diameter: 10 mm) by a lathe. The effect of residual stress on the corrosion behavior of the USRP-treated alloy was determined by performing a residual stress relaxation process on circular cross-sectional fatigue specimens (gauge length: 35 mm, diameter: 10 mm). This process was conducted using 0.1 Hz sine wave with stress ratio = −1 under a maximum stress of 350 MPa and was run for 10 cycles on a fatigue testing machine (SDS-100, Changchun Research Institute for Mechanical Science Co., Ltd., Changchun, China).

A schematic of the USRP device (HK30G, Shandong Huawin Electrical & Mechanical Technology Co., Ltd., Jinan, China) is shown in Figure 1, and the processing parameters are listed in Table 2. The samples without USRP treatment were referred to as BM. When the tungsten carbide/cobalt (WC/Co) ball slid from one end of the sample surface to the other, the sample underwent a single pass of USRP treatment, which was referred to as UR1. UR12-P refers to a sample that has undergone 12 rounds of USRP and removal of a 40-μm-thick surface layer. This removal was aimed at eliminating the micro-damage caused by the 12-pass URSP treatment. The removal method involved a combined treatment of chemical stripping (corrosive solution: 200 g/L sodium hydroxide + 20 g/L sodium sulfide + 35 mL/L triethanolamine) [25] polishing, i.e., the stripped sample was rotated on a lathe at 100 rpm and polished with #3000 silicon carbide sandpaper. After the process of residual stress relaxation, the UR1 and UR12-P samples were referred to as UR1-R and UR12-PR, respectively. Microstructural characterization, immersion testing, and electrochemical measurements were performed on cylindrical samples (length: 15 mm) cut from the aforementioned samples. Three parallel specimens (of the same type) were chosen for each measurement, and the average of the data was taken.

### 2.2. Microstructural Characterization

The surface morphologies of the samples with and without USRP treatment were observed by means of scanning electron microscopy (SEM; Vega II, Tescan, Kohoutovice, Czech Republic). In addition, the average surface roughness (*S*a) was determined via confocal laser scanning microscopy (CLSM; LSM 700, Carl Zeiss AG, Oberkochen, Germany). The surface microstructure of different samples was characterized by means of transmission electron microscopy (TEM; Tecnai G^2^ F20, FEI, Hillsboro, OR, USA) performed at an accelerating voltage of 200 kV. Samples for TEM observation were prepared via single-sided ion-beam milling.

The surface residual stress of the untreated and USRP-treated samples was measured via an X-ray stress analyzer (MSF-3M instrument with CrKα radiation, Rigaku, Tokyo, Japan). In accordance with the sin^2^*ψ* method, the normal angles of *ψ* were 0°, 18.40°, 26.60°, 33.20°, 39.20°, and 45.00°. Besides, the 139.3° diffraction angle and {311} diffraction peak of the aluminum alloy were selected as measurement parameters. The crystal structure of each sample was analyzed via X-ray diffraction (D/max 2500 instrument with CuKα radiation, Rigaku, Tokyo, Japan) performed at a 2θ scan rate of 8°/min.

### 2.3. Immersion Testing

In accordance with ASTM standard G110-92 [26], the test solution was prepared by diluting 57 grams of NaCl and 10 mL of H_2_O_2_ with 1 L of distilled water. The glass test vessel could hold 20 mL of test solution per square centimeter of sample surface area. The cylindrical samples (including BM, UR1, UR1-R, UR12-P, and UR12-PR) were immersed in the test solution for 24 h at 30 ± 3 °C. After immersion, each sample was rinsed with reagent water and allowed to dry, and the corresponding cross-sections were then etched with Keller’s reagent. The surface and cross-sectional morphologies of the tested samples were observed via SEM. The pitting area percentage and the corrosion depth were calculated from each of the resulting SEM images (at least three images were obtained for each sample) via Image-Pro Plus software (version 6.0, Media Cybernetics, Inc., Rockville, MD, USA). Furthermore, the chemical compositions of the corrosion products and oxide layer were determined via energy-dispersive X-ray spectroscopy (EDS; INCA Energy 350 EDX analyzer, Oxford Instruments, Oxfordshire, UK).

### 2.4. Electrochemical Measurements

Using a PARSTAT 2273 electrochemical station (AMETEK, Inc., Berwyn, PA, USA) connected to a three-electrode cell, electrochemical measurements were performed automatically under the control of PowerSuite software (version 2.47, Princeton Applied Research, Oak Ridge, TN, USA). All the measurements were conducted in the NaCl + H_2_O_2_ solution. A thin platinum foil and a saturated calomel electrode (SCE) and were used as the auxiliary and reference electrodes, respectively. The untreated and USRP-treated cylindrical samples were fabricated into working electrodes via the insertion of insulated copper wires, leaving an exposed surface area of 0.5 cm^2^. The potentials mentioned in the present work were all measured with respect to the potential of SCE.

Open circuit potential (OCP)-time curves were obtained for the aforementioned samples. When the OCP was stable, a potentiodynamic polarization scan starting at −250 mV vs. OCP, and ending at 250 mV (scan rate: 0.5 mV/s) was performed on each sample.

When the 7B50 Al alloy is exposed to air, a passive film is naturally formed on its surface, which is closely related to the initial stage of the corrosion behavior of this alloy in the immersion test solution. Therefore, the Mott–Schottky curve was measured and used to analyze the passive film in the case of the untreated and USRP-treated samples. The potential was scanned from −250 mV vs. OCP to 500 mV (scanning interval: 10 mV). Moreover, an alternating current signal with a frequency of 1000 Hz and an amplitude of 5 mV was superimposed on the scanning potential. The impedance value was obtained from the PowerSuite software, and the corresponding capacitance (*C*) at each applied potential was calculated as follows [27]:*C* = −1/(2π·*f*·*Z*_im_)(1)
where *f* = 1000 Hz and *Z*_im_ is the imaginary part of the impedance. The value of *C* (F/cm^2^) is normalized for the area.

The EIS measurements were conducted at OCP with a sinusoidal 10 mV perturbation signal and frequency ranging from 100 kHz to 0.1 Hz. The data from each sample were recorded at 2 h, 6 h, 12 h, and 24 h, and then analyzed by ZsimpWin software (version 3.60, EChem Software, Ann Arbor, MI, USA).

## 3. Results

### 3.1. Surface Morphology and Microstructure

Scanning electron micrographs showing the surface of the untreated and USRP-treated 7B50 aluminum alloy sample are demonstrated in Figure 2a–c. Longitudinal machining marks occurred on the BM sample surface, but were almost absent from the UR1 sample. Similarly, circumferential polishing marks were observed on the UR12-P sample surface. The inset of each image shows the corresponding surface CLSM result. The mean surface roughness (Sa) values of the BM, UR1, and UR12-P samples were 0.698 ± 0.009 μm, 0.186 ± 0.030 μm, and 0.269 ± 0.037 μm, respectively. Compared with the polishing treatment, the one-pass USRP treatment yielded a significantly greater reduction in the surface roughness of 7B50 aluminum alloy. The influence of USRP on the surface morphology of the alloy has been discussed in our previous study [25]. For example, the surface roughness and damage increased with the number of USRP treatment.

The TEM images in Figure 2d–f show the surface microstructure of the samples with and without the USRP treatment. The semi-coherent matrix precipitates (η’-MgZn_2_) [28] and the coherent G.P. zones [28] were uniformly scattered in the grains of the untreated sample (Figure 2d). Additionally, the disconnected grain boundary precipitates (η-MgZn_2_) and the precipitate free zone (PFZ) were clearly observed. For the UR1 sample, some dislocation tangles (DTs) and dislocation walls (DWs) appeared in grains due to USRP-introduced plastic deformation (Figure 2e). However, the matrix precipitates and grain boundary structure of UR1 the sample were basically the same as those of the BM sample. The TEM image and corresponding selected area electron diffraction (SAED) pattern of the UR12-P sample shown in Figure 2f reveals that coarse aluminum grains were sub-divided into fine grains with high angle boundaries. Moreover, the η’-MgZn_2_ phase, η-MgZn_2_ phase, and G.P. zones have almost disappeared from this sample. The mean grain size of the surface layer comprising the UR12-P sample is ~67 nm, as revealed by statistical analysis performed in our previous study [24], i.e., a nanocrystalline structure has formed on the surface of the UR12-P sample. For the aluminum alloy subjected to severe plastic deformation, the major mechanism of grain refinement occurs through accommodation of the high plastic strain via grain sub-division into subgrains [14].

### 3.2. Compressive Residual Stress and XRD

Figure 3a shows that the USRP treatment introduced axial residual stress to the surface of the samples, where the negative values represent the compressive residual stress. The samples may be listed in ascending order of the surface compressive residual stress, i.e., BM < UR1-R < UR12-PR < UR12-P < UR1. The corresponding stresses are 38.65 ± 11.48, 121.63 ± 8.04, 135.89 ± 9.76, 171.70 ± 6.32, and 219.73 ± 11.13 MPa, respectively. The compressive residual stress value of UR1-R was 44.65% lower than that of the UR1 sample, whereas the value of UR12-PR was only 20.87% lower than that of the corresponding unrelaxed sample. This result indicated that the compressive residual stress at the surface of the UR12-P sample is more stable than that of the UR1 sample.

In general, {111} <112>, {111} <110>, {001} <110>, and {112} <110> are the major slip systems of the Al alloys subjected to shear deformation [29]. At least two sets of these slip systems associated with the 7B50-T7751 Al alloy were activated to produce the required strain during the USRP treatment. The relative intensity of the diffraction peaks corresponding to the {111} planes increased gradually following the one-pass and 12-pass USRP treatments, as shown in Figure 3b. This indicated an increase in the number of these diffraction planes comprising the surface of the USRP-treated samples. Verdan et al. [30] reported a similar structural change in the aluminum surface due to the mechanical effect. Additionally, the XRD peak broadening and left-shifting of the samples following the 12-pass USRP treatment were observed, especially for the diffraction peaks corresponding to the (222) planes. The microstrain and grain refinement caused by the USRP treatment resulted mainly from the XRD peak broadening. Owing to the dissolution of the η’-MgZn_2_, η-MgZn_2_ phases, and G.P. zones, the increment in the interplanar distance of aluminum resulted in XRD peak left-shifting. Similar results have been obtained for 7150 Al alloy after ultrasonic shot peening treatment [12].

### 3.3. Immersion Test

The SEM images in Figure 4a–e show the typical surface morphologies of untreated and USRP-treated samples after immersion tests. The micrograph of the BM sample revealed a widespread pitting attack of the sample surface, but this condition was improved after USRP treatment, as evidenced by a decrease in the pitting area. As shown in Figure 4f, the samples may be listed in ascending order of average pitting area percentage, i.e., UR12-P < UR12-PR < UR1 < UR1-R < BM. The average pitting area percentage (3.67 ± 0.89%, 3.87 ± 1.59%, 6.34 ± 2.87%, and 13.06 ± 3.81%) of the UR12-P, UR12-PR, UR1, and UR1-R samples are 80.16%, 79.08%, 65.73%, and 29.41% lower, respectively, than that (18.50 ± 0.71%) of the BM sample. This comparison indicated that, irrespective of the residual stress relaxation process, the pitting area of the USRP-treated samples with a nanocrystalline surface is always lower than that of the untreated sample. In contrast, the pitting area percentage of the UR1-R sample was higher than that of the UR1 sample. This confirmed that the compressive residual stress has a significant effect on the pitting sensitivity of the one-pass USRP-treated samples without a nanocrystalline surface.

Figure 5 shows the SEM/EDS results of the surface corrosion products and oxide layer for the untreated sample, UR1 sample, and UR12-P sample. A layer of corrosion products with a muddy pattern can be clearly observed on the surface of each sample. The overall reaction of the pitting corrosion on aluminum alloys in a neutral aqueous medium is given as: 2Al + 3H_2_O + 3/2O_2_→2Al(OH)_3_ [1]. The unstable aluminum hydroxide gel Al(OH)_3_ crystallizes and forms the monoclinic trihydrate Al_2_O_3_·3H_2_O [4]. Consider the aforementioned reactions and the results shown in EDS spectra 1, 3, and 5 (an atomic ratio close to 1:3 for Al and O, excluding the oxygen in Zn(OH)_2_ and Mg(OH)_2_). These reactions and results confirmed that the surface corrosion product of untreated and USRP-treated 7B50 aluminum alloy samples is Al_2_O_3_·3H_2_O. The USRP treatment had no influence on the composition of the corrosion products in the external layer. However, EDS spectra 2, 4, and 6 of the regions beneath the products showed that the composition of the thin oxide layer, in contact with the alloy, had changed. The atomic content of oxygen of the thin oxide layer in the UR12-P sample was about twice that of the BM and UR1 samples. This indicated that the nanocrystalline surface may promote the formation of Al_2_O_3_·3H_2_O (rather than amorphous Al_2_O_3_ [1]) in the oxide layer during 24-h immersion testing. Moreover, Al_2_O_3_·3H_2_O formation is beneficial for improving the corrosion resistance. Similarly, Trdan [11] evaluated the corrosion resistance of laser shock peened AA6082-T651 aluminum alloy after 24 h of exposure to 0.6 M NaCl. He found that this treatment transformed amorphous Al_2_O_3_ into a more stable oxide form in the surface film, thereby forming an effective anti-corrosion barrier.

The SEM images presented in Figure 6a–e show representative cross-sectional morphologies of untreated and USRP-treated samples after 24-h immersion tests. Deep corrosion pits with large openings were observed in the surface layer of the BM sample, and slight IGC occurred at the bottom of these pits (see Figure 6a,b). As shown in Figure 6c,e, the opening size and depth of the pits decreased and the IGC was significantly inhibited after the USRP treatment, especially for the UR12-P sample. The opening size and depth of the pits in the UR1-R sample increased relative to those of the UR1 sample, and IGC occurred beneath the pits (Figure 6d). The UR12-PR sample exhibited low susceptibility to pitting corrosion and IGC (Figure 6f). Furthermore, bright particles, confirmed as Fe-rich intermetallic particles in our previous study, were observed in Figure 6a–g [31]. As shown in the inset of Figure 6g, the high strains introduced by 12-pass USRP led to the formation of a lamellar structure or ribbon grains in the surface layer. This structure broke up the bright particles aligned in the layer, i.e., Fe-rich intermetallic particles were refined in the severely deformed layer.

To further investigate the influence of USRP treatment on the pitting attack of 7B50-T7751 Al alloy, the corrosion depth was obtained from 35 different corrosion pits in each sample and statistical analysis was then performed. Figure 6h shows the box plots of the depths determined for different samples (see Table 3 for the detailed statistics). The results revealed that the mean corrosion depths of the UR12-P, UR12-PR, UR1, and UR1-R samples were 55.64%, 47.96%, 29.14%, and 15.42% lower, respectively, than those of the BM sample. Furthermore, other statistical parameters (including the minimum, first quartile, median, third quartile, and maximum) describing the depth of the UR12-P sample were lower than those of the other samples. The statistical results suggested that the lowest pitting degree occurred for the 7B50 aluminum alloy with a nanocrystalline surface layer, where the alloy was soaked for 24 h in the test solution.

### 3.4. OCP and Polarization

Figure 7a shows the OCP variation of different samples in the NaCl + H_2_O_2_ solution with testing time. As shown in the figure, the OCP values of each sample stabilized after 10 min of testing. The samples can be listed in ascending order of the OCP values, i.e., BM < UR1-R < UR1 < UR12-PR < UR12-P, corresponding to values of ~−755 mV, −747 mV, −745 mV, −738 mV, and −737 mV, respectively. The OCP values of the samples with a nanocrystalline surface layer were more positive than those of the other samples. Figure 7b displays the polarization curves of different samples immersed for 15 min in the solution. The corrosion current density (*i*_corr_) is obtained through extrapolation of the Tafel slope, and the corresponding corrosion rate (CR; μm/year) can then be calculated from Faraday’s law, which is given as follows [32]:CR = 10.89·*i*_corr_(2)
where the unit of *i*_corr_ is μA/cm^2^.

The values of the free corrosion potential (*E*_corr_), *i*_corr_, and CR obtained for the untreated and USRP-treated samples are listed in Table 4. As shown in the table, the *E*_corr_ values of 12-pass USRP-treated samples were more positive than the *E*_corr_ values of the other samples, as in the case of OCP shifting. The samples may be listed in ascending order of *i*_corr_, i.e., UR12-P < UR12-PR < UR1 < UR1-R < BM. Furthermore, the CR values of the UR12-P, UR12-PR, UR1, and UR1-R samples were 39.00%, 36.67%, 23.76%, and 2.16% lower than that of the BM sample.

### 3.5. Mott–Schottky Analysis

In accordance with the Mott–Schottky theory, the space charge capacitance (*C_SC_*) of a semiconductor with an electrode potential (*E*) is given as follows [33]:(3)$$C_{SC}^{-2}=\frac{2}{\varepsilon \varepsilon_{0}qN_{D}A^{2}}\left(E-E_{fb}-\frac{kT}{q}\right)$$
(4)$$C_{SC}^{-2}=\frac{-2}{\varepsilon \varepsilon_{0}qN_{A}A^{2}}\left(E-E_{fb}-\frac{kT}{q}\right)$$
where *ε* is the relative dielectric constant of a semiconductor (this value is 10 for the oxide film on aluminum alloys [34]), *ε*_0_ is the permittivity of vacuum (8.854 × 10^−14^ F cm^−1^), and *N_D_* and *N_A_* are the donor density and acceptor density, respectively. *A* is the interfacial surface area, *q* is the electronic charge (1.602 × 10^−19^ C), *E_fb_* is the flat-band potential, *k* is the Boltzmann constant (1.38 × 10^−23^ J K^−1^), and *T* represents the Kelvin temperature. For an n-type semiconductor, the slope of the linear segment comprising the Mott–Schottky plot is positive. A negative slope is obtained for a p-type semiconductor. Additionally, for a semiconducting oxide electrolyte interface, the measured capacitance *C* is an equivalent capacitance of two serialized capacitors (the Helmholtz layer capacitance *C_H_* and the space charge capacitance *C_SC_*), which can be expressed as [35]: 1/*C* = 1/*C_H_* + 1/*C_SC_*. The *C_H_* on the solution side of the metal-solution interface is ∼50 times larger than the *C_SC_* (which is almost negligible), and hence, *C* ≈ *C_SC_* [36].

Based on the above theory, the *C**^−2^*−*E* plots (Figure 8a) were obtained for different samples immersed for 15 min. The corresponding *E_fb_* values were determined by extrapolating the linear segment to C^−2^ = 0 (see Table 5). The *E_fb_* of passive film on the 12-pass USRP-treated samples was more positive than that of the untreated sample. Negative slopes occurred in the region from −0.75 V_SCE_ to −0.6 V_SCE_, consistent with the p-type semiconductor behavior of the film. This behavior may be attributed to a preponderance of metallic cation vacancies, i.e., the primary charge carriers in the passive film are metallic cation vacancies. At the same time, the oxygen produced by the decomposition of H_2_O_2_ and H_2_O molecules can easily enter the oxide film and fills the oxygen vacancies [36]. Furthermore, Martin et al. [35] reported that the space charge layer thickness (*d*) at *E*_corr_ can be calculated from *d* = *ε·ε*_0_/*C_SC_*, where *C_SC_* ≈ *C*. This thickness is a small fraction (~1/10) of the passive film thickness (*δ*), i.e., *δ* ≈ 10*d*. The *δ* of different samples can be estimated from this relation (see Table 5). The *δ* values of the UR1, UR1-R, UR12-P, and UR12-PR samples were significantly (i.e., 28.88%, 14.66%, 55.17%, and 47.41%, respectively) higher than that of the BM sample. In other words, the thickest passive film occurred on the UR12-P sample during the initial stage of corrosion (soaking in test solution for 15 min). Figure 8b shows the acceptor concentrations calculated from the Mott–Schottky plots for *E* = *E*_corr_. For the BM sample, the acceptor density of the oxide film (23.68 × 10^20^ cm^−3^) was higher than those determined for the USRP-treated samples. Additionally, the acceptor densities of the UR1-R and UR12-PR samples were 23.10% and 17.30% higher, respectively, than those of the corresponding residual stress unrelaxed samples.

### 3.6. EIS

Figure 9 shows the typical EIS results obtained for the untreated 7B50 aluminum alloy immersed in the test solution for different times. As shown in the figure, Nyquist plots with similar shapes were obtained in the complex plane. All the plots consisted of a small capacitive semicircle, a large capacitive semicircle, and an inductive circular arc. The radius of the large semicircle increased significantly with increasing immersion time. Two crests for the positive value and a trough for the negative value in each phase angle curve (Bode plot) indicated the interaction of three time constants [37]. The number of time constants corresponded to the number of electronic components. One time constant occurred in each frequency domain (i.e., high (10^2^–10^3^ Hz)-, medium (1–10 Hz)-, and low (0.1–0.2 Hz)-frequency domains). At a frequency of ~0.4 Hz, the impedance modulus (|Z|) reached a maximum value for all immersion times. Thereafter, |Z| decreased at frequency values of <0.4 Hz and the phase angle shifted to negative values, owing to the inductive behavior occurring in the low-frequency domain.

Figure 10 shows the impedance response of the UR1 and UR1-R samples at different immersion times. When the immersion time was less than 6 h, the shapes of the Nyquist and Bode plots for these two types of samples were similar to those of the BM sample. After soaking for 12 h and 24 h, the inductive circular arcs disappeared in the case of the UR1 sample, but persisted in the case of the UR1-R sample. Generally, the inductive circular arc results from any of the three following cases: (1) the electrolyte contains a reducible inhibitor, (2) the intermediate product formed during the electrode process, (3) the initiation of pitting on the electrode surface [38]. In any case, except for the electrode potential, the electrode process is affected by multiple state variables. Examination of the corrosion surface morphology (Figure 4) revealed many large pits on the surface of the BM and UR1-R samples, indicating that the inductive circular arc is correlated with the pitting corrosion.

The Nyquist and Bode plots of the UR1, UR12-P, and UR12-PR samples exhibited similar trends (see Figure 11). However, the inductive circular arcs are absent from the plots of the UR12-P and UR12-PR samples after 2-h immersion in the test solution. A comparison of Figure 9, Figure 10 and Figure 11 revealed that, for the same immersion time, the samples may be listed in ascending order of the radius increment characterizing the medium-frequency capacitive semicircle: BM < UR1-R < UR1 < UR12-PR < UR12-P. Furthermore, the |Z| value of UR12-P was always higher than those of other samples during immersion testing, indicating the high corrosion resistance of this sample.

As shown in Figure 12a,b, the EIS results are accurately analyzed via equivalent circuit 1 *R*_s_(*Q*_c_(*R*_po_(*Q*_dl_
*R*_ct_(*L R*_L_)))) and equivalent circuit 2 *R*_s_(*Q*_c_(*R*_po_(*Q*_dl_
*R*_ct_))). These two circuits are suitable for investigating the corrosion behavior of aluminum alloys [32,39,40]. The physical interpretation of the aforementioned equivalent circuit elements is as follows: *R*_s_, *R*_po_, *R*_ct_, and *R*_L_ are the solution resistance, pore resistance, charge-transfer resistance, and equivalent resistance introduced by the inductance, respectively. *Q*_c_ and *Q*_dl_ represent the constant phase elements (CPE), i.e., imperfect capacitances, of the porous protective oxide film and the double charge layer, respectively. The admittance of a CPE is defined as *Y*_CPE_ = *Y*_0_ (*jω*)^n^, where *Y*_0_ is the CPE coefficient, *j*^2^ = −1, and n is the CPE exponent [39]. *L* is the inductance. The high-frequency time constant (*Q*_c_, *R*_po_) represents reactions at the porous oxide film–electrolyte interface. Similarly, the medium-frequency time constant (*Q*_dl_, *R*_ct_) represents reactions at the dense oxide film–electrolyte interface, and the low-frequency time constant (*L*, *R*_L_) is correlated with the development of pitting corrosion. The good fitting results (Chi-square values on the order of 10^−4^ were obtained) of equivalent circuits for the untreated and USRP-treated samples immersed for different times are shown in Figure 9, Figure 10 and Figure 11. Table 6 shows the fitted electrochemical parameters of the samples immersed for 2 h. As shown in the table, the highest *R*_ct_ value and the lowest value were obtained for the UR12-P sample and the BM sample, respectively.

Figure 13 shows the polarization resistance *R*_p_ of the untreated and USRP-treated sample as a function of the immersion time (*R*_p_ = *R*_po_ + *R*_ct_·*R*_L_/(*R*_ct_ + *R*_L_) and *R*_p_ = *R*_po_ + *R*_ct_ for circuits 1 and 2, respectively). As shown in the figure, the *R*_p_ values increased gradually with an increasing immersion time of each sample, and the *R*_p_ values of the USRP-treated samples were higher than that of the BM sample. *R*_p_ values of 206.12 Ω cm^2^ and 175.24 Ω cm^2^ were obtained for the 24-h-immersed UR12-P sample and UR1 sample. This suggested that the 7B50 aluminum alloy with a nanocrystalline surface layer has a high *R*_p_ value. Furthermore, the *R*_p_ values of the UR1-R and UR12-PR samples were lower than those of the corresponding unrelaxed samples, indicating that the *R*_p_ value is also correlated with the compressive residual stress.

## 4. Discussion

Owing to the discontinuous η-MgZn_2_ precipitates at the grain boundary [41,42], 7B50-T7751 Al alloy exhibited low susceptibility to IGC in the NaCl + H_2_O_2_ solution. However, apparent pitting corrosion occurred on the surface of the alloy, owing mainly to the reaction between the cathodic Fe-rich intermetallic particles and the nearby aluminum matrix [43]. The pitting corrosion resistance of the 7B50 Al alloy was improved significantly due to the ultrasonic surface rolling process, which had a considerable effect on pitting initiation and propagation.

### 4.1. Initiation of Pitting

Pits are often initiated at a surface, where chemical or physical heterogeneity (including dislocations, inclusions, second phase particles, mechanical damage, or flaws) occurs [44]. After the one-pass and 12-pass USRP treatment, the Sa values of the samples decreased by 73.35% and 61.46%, respectively, relative to that of the untreated sample. The surface roughness and sensitivity to pitting attack of the treated samples are lower than those of the untreated samples; this may be associated with the harder penetration of water molecules, oxygen, and chloride ions into isolated regions of inhomogeneities (intermetallics, passive film defects, pores, etc.) [32]. Furthermore, the 12-pass USRP treatment resulted in surface nanocrystallization of the 7B50 aluminum alloy. Moreover, partial dissolution of the precipitates (distributed in the matrix and at the grain boundary) and the refinement of Fe-rich intermetallic particles yielded a surface layer with a relatively homogeneous microstructure (see Figure 2 and Figure 6). This layer was conducive to the inhibition of pitting initiation, consistent with Ralston’s findings [45]. That is, the fine intermetallic particles did not lead to a breakdown in the passive film stability and, hence, pit formation was hindered. Additionally, the enhanced re-passivation during pit formation was facilitated by the high surface chemical activity induced by the grain refinement, consistent with the findings of a previous study [46].

The USRP treatment introduced a high surface compressive residual stress in the UR1 and UR12-P samples (see Figure 3a), which was beneficial for restraining the initiation of localized corrosion in aluminum alloys [47,48]. Meanwhile, USRP led to an increase in the number of {111} diffraction planes on the surface of each treated sample, especially the 12-pass USRP-treated samples (see Figure 3b). The crystal planes of aluminum can be listed in descending order of the surface energy, i.e., {111} < {100} < {110} < high-index crystal planes [49]. Therefore, the corrosion resistance of the 7B50 Al alloy will increase with an increasing number of {111} crystal planes. Similarly, Wang et al. [50] found that the AA 6082 alloy with a shear orientation (preferred {100} and {111} crystal planes) exhibited a high corrosion resistance in a 3.5% NaCl solution.

The OCP-time curves of the USRP-treated samples were consistent with a hypothesis proposed by several scholars [32,51], i.e., a dense and easily passivized oxide film formed after severe plastic deformation treatment leads to an anodic shift in the OCP. The Mott–Schottky analysis confirmed that, compared with the passive film on the BM sample, the film formed on the USRP-treated samples was thicker and characterized by a lower acceptor density. According to the point defect model [52], the movement of cation vacancies and oxygen vacancies contributed to the formation and breakdown of the passive film. The low acceptor density indicated that the USRP treatment may reduce the dissolution of metallic cations in the film, thereby retarding the breakdown of the film. This is beneficial for inhibiting the initiation of pitting corrosion. Lv et al. [27] reported similar results. Considering the *δ* and the acceptor density characteristics of the passive film, the compressive residual stress had a significantly stronger effect on the film characteristics of the one-pass USRP-treated samples than on the film characteristics of the 12-pass USRP-treated samples (see Table 5 and Figure 8b).

### 4.2. Propagation of Pitting

The morphologies of the corrosion pits (see Figure 4, Figure 5 and Figure 6) were compared. This comparison revealed that the pits in the BM, UR1, and UR1-R samples expanded toward the interior of these samples, whereas those in the 12-pass USRP-treated samples grew mainly along the circumferential direction. In other words, the propagation of the pitting attack was based on the configuration of aluminum grains in the surface layer. The factors affecting the morphology of pits formed in aluminum alloys have been reviewed by Soltis [53]. Pit propagation in the one-pass USRP-treated samples was apparently affected by the compressive residual stress, i.e., deep pits and IGC were observed in the UR1 sample subjected to the residual stress relaxation (UR1-R sample). The occurrence of an inductive circular arc in the impedance response of the UR1-R sample further showed that this stress played a significant role in inhibiting pit propagation (see Figure 10). Liu et al. [54] proposed that a normal compressive stress applied to the IGC regions can significantly reduce the number of active sites and retard the growth rates of these sites.

The impedance responses of the untreated and USRP-treated samples were closely correlated with the occurrence of pitting (see Figure 9, Figure 10 and Figure 11). The disappearance of the inductive circular arc after a 2-h immersion of the UR12-P and UR12-PR samples with a nanocrystalline surface implied that the propagation of pitting corrosion was hindered (see Figure 11). This may be correlated with the grain refinement of these samples (see Figure 6). Moreover, the low corrosion rates (Table 4) of these samples were consistent with the corresponding average pitting area percentage and corrosion depth. Ralston et al. [55] reported that the corrosion rates decreased with decreasing grain size in systems that exhibit some level of passivity.

The *R*_ct_ value of the BM sample was lower than that of the USRP-treated samples (see Table 6). This indicated that a large surface area of the sample came into contact with the electrolyte due to the formation of large stable pits [3,56]. Considering the relationship between *R*_ct_ and *R*_p_, the degree of pitting should increase with decreasing *R*_p_ value. Furthermore, the continuous deposition of corrosion products reduced the contact area between the samples and the electrolyte during the soaking process, resulting in an increase in the *R*_p_ value of each sample (see Figure 13). The *R*_p_ values of the UR12-P and UR12-PR samples were higher than those of the UR1 and UR1-R samples. This was attributed to the surface microstructure homogenization caused by the 12-pass USRP treatment, which reduced the surface electrochemical heterogeneity of the sample. This is consistent with the findings of Sun et al. [6], who reported that the corrosion resistance of ultrasonic shot-peened AA2024 can be improved by obtaining a surface layer with a homogeneous microstructure.

In summary, for one-pass USRP-treated 7B50 aluminum alloy samples, the introduced compressive residual stress played a key role in the improvement of the pitting corrosion resistance. For the 12-pass USRP-treated samples, the increased resistance resulted mainly from grain refinement and homogenization of the surface layer microstructure (i.e., the residual stress played a secondary role).

## 5. Conclusions

1. The 7B50-T7751 aluminum alloy exhibited low susceptibility to intergranular corrosion in the NaCl + H_2_O_2_ solution, but underwent considerable pitting corrosion (average pitting area percentage: 18.50 ± 0.71%, mean corrosion depth: 147.2 ± 69.8 μm). This alloy was characterized by a high corrosion rate, thin passive film, low polarization resistance, and extremely high acceptor concentration of the passive film, as revealed by the electrochemical testing.

2. Owing to a combination of several factors, both one-pass and 12-pass USRP treatment can yield a significant improvement in the corrosion resistance of 7B50-T7751 aluminum alloy. The average pitting area percentages of UR1 and UR12-P samples were 6.34 ± 2.87% and 3.67 ± 0.89%, respectively, and the corresponding mean corrosion depths were 104.3 ± 59.1 μm and 65.3 ± 38.5 μm, i.e., 12-pass USRP treatment yielded greater improvement than the one-pass treatment.

3. Compared with the untreated sample, the UR1 and UR1-R samples were characterized by lower (65.73% and 29.41%) pitting area percentage and lower mean corrosion depth (29.14% and 15.42%, respectively). The electrochemical test results also revealed that the corrosion resistance of the one-pass USRP-treated samples had improved. This improvement resulted mainly from the introduced compressive residual stress.

4. A comparison of all the samples revealed that the UR12-P sample is characterized by the: minimum average pitting area percentage, lowest mean corrosion depth, minimum corrosion rate, thickest passive film, maximum polarization resistance in the test solution, and the lowest acceptor concentration of the passive film. The residual stress relaxation process had little effect on the above results. Grain refinement and surface layer microstructural homogenization played the dominant role in improving the corrosion resistance of the 12-pass USRP-treated samples, and the induced compressive residual stress played a secondary role.

## Figures and Tables

**Figure 1 materials-13-00738-f001:**
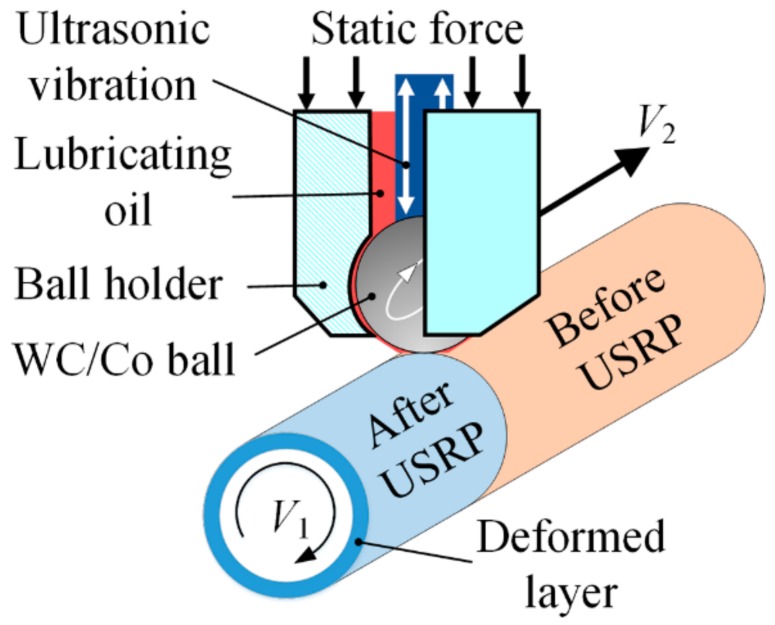
Schematic of the ultrasonic surface rolling process.

**Figure 2 materials-13-00738-f002:**
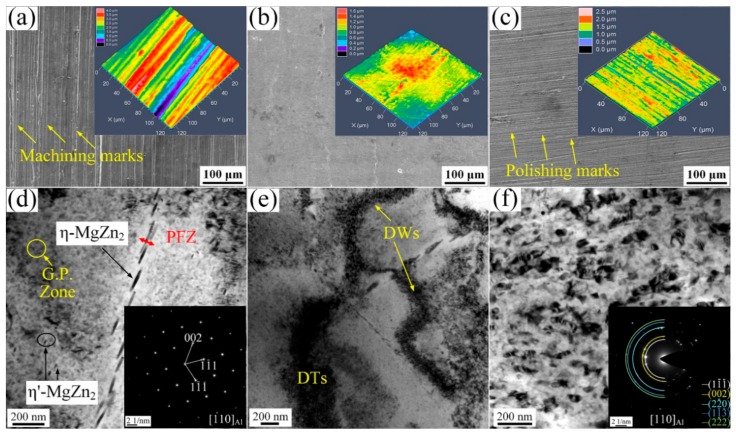
Surface SEM secondary electron images of different samples: (**a**) untreated sample (BM), (**b**) one-pass USRP treatment (UR1), and (**c**) 12-pass USRP treatment (UR12-P). Insets display the corresponding CLSM images. TEM images showing the microstructure in the surface of (**d**) BM, (**e**) UR1, and (**f**) UR12-P. Insets show the corresponding SAED patterns.

**Figure 3 materials-13-00738-f003:**
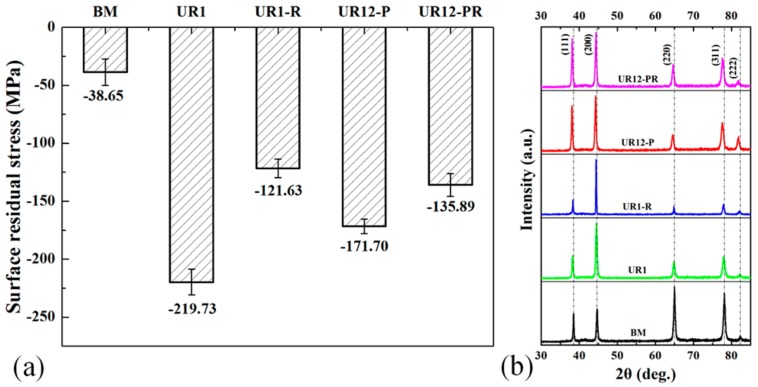
(**a**) Surface residual stress and (**b**) XRD patterns of different samples.

**Figure 4 materials-13-00738-f004:**
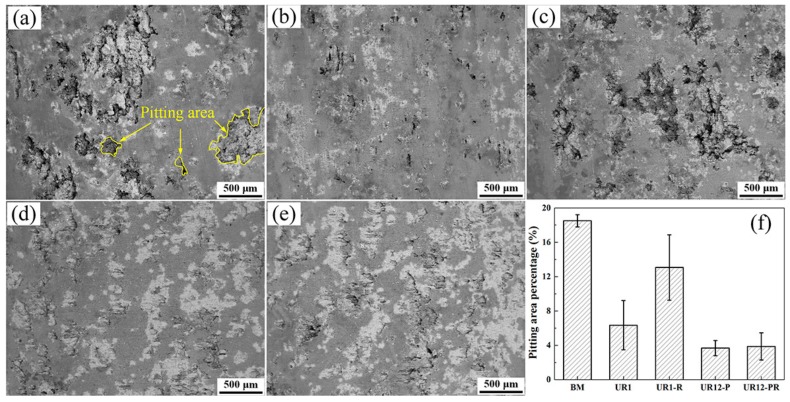
Representative surface SEM backscattered electron images of 7B50 aluminum alloy samples subjected to immersion tests: (**a**) untreated sample, (**b**) UR1, (**c**) UR1 after residual stress relaxation (UR1-R), (**d**) UR12-P, and (**e**) UR12-P after residual stress relaxation (UR12-PR). (**f**) Pitting area percentage of these samples.

**Figure 5 materials-13-00738-f005:**
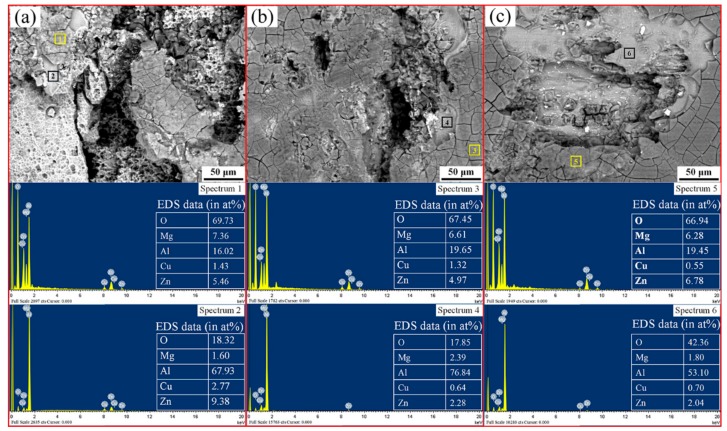
SEM/EDS analysis of the surface corrosion products and oxide layer of the samples after the immersion test: (**a**) BM, (**b**) UR1, and (**c**) UR12-P.

**Figure 6 materials-13-00738-f006:**
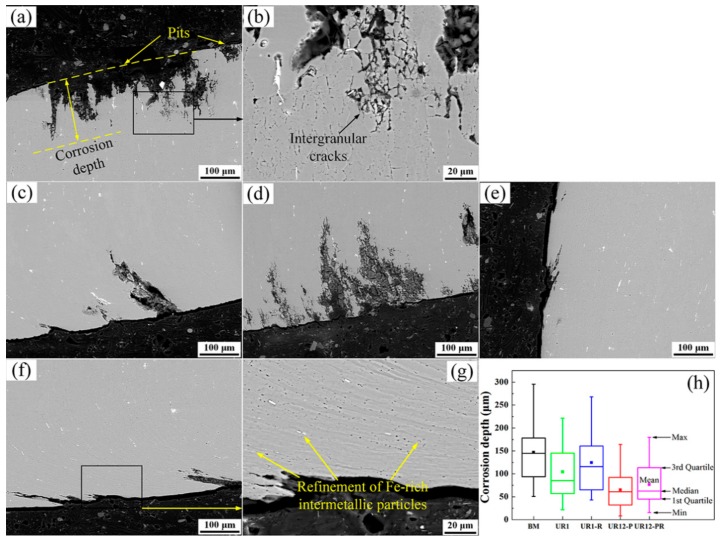
Typical cross-sectional SEM backscattered electron images of 7B50 aluminum alloy samples subjected to immersion tests: (**a**,**b**) untreated sample, (**c**) UR1, (**d**) UR1-R, (**e**) UR12-P, and (**f**,**g**) UR12-PR. (**h**) Corrosion depth statistics of these samples.

**Figure 7 materials-13-00738-f007:**
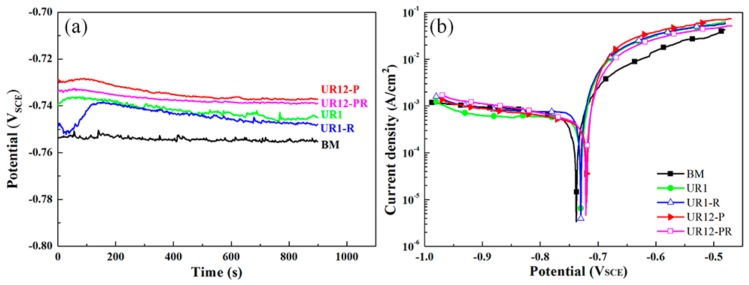
The open circuit potential (OCP)-time curves (**a**) and the polarization curves (**b**) of different samples immersed in the NaCl + H_2_O_2_ solution.

**Figure 8 materials-13-00738-f008:**
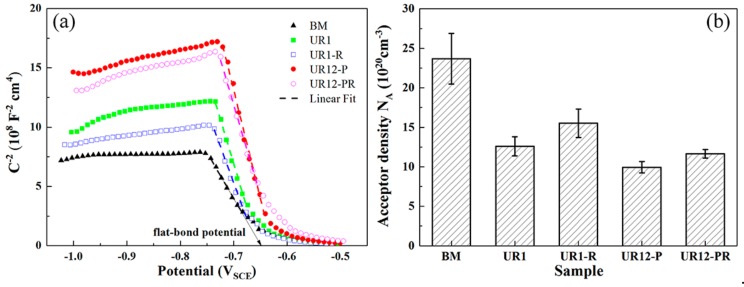
(**a**) Mott–Schottky plots of different samples after 15 min stabilization in the test solution and (**b**) the calculated acceptor densities.

**Figure 9 materials-13-00738-f009:**
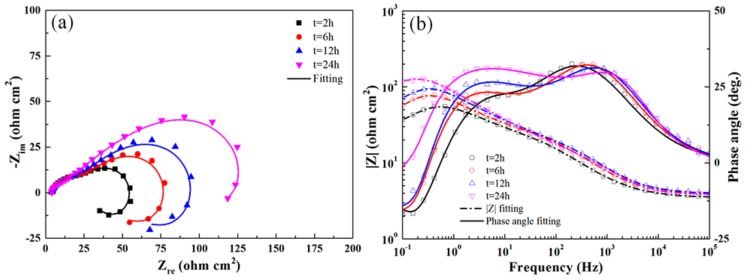
Nyquist (**a**) and Bode (**b**) plots obtained for the BM sample immersed for different times in the test solution at OCP.

**Figure 10 materials-13-00738-f010:**
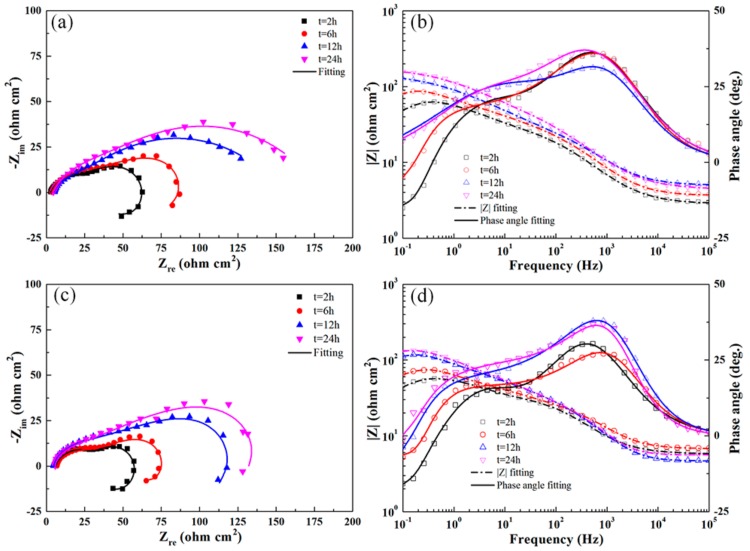
Nyquist (**a**,**c**) and Bode (**b**,**d**) plots obtained for UR1 and UR1-R samples immersed for different times in the test solution at OCP.

**Figure 11 materials-13-00738-f011:**
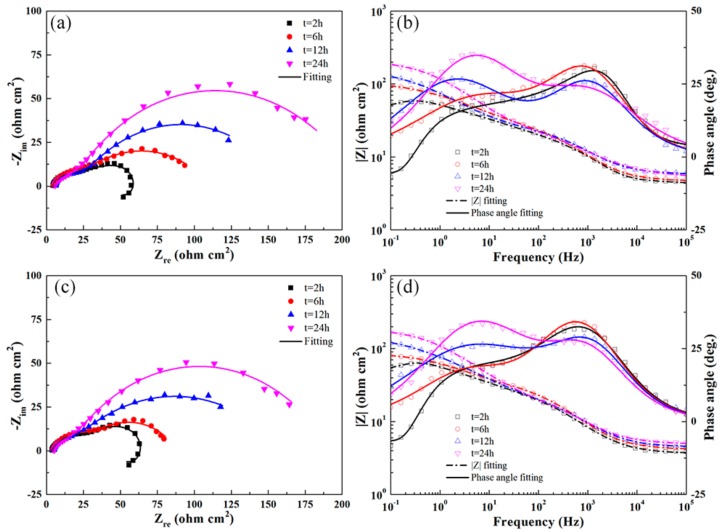
Nyquist (**a**,**c**) and Bode (**b**,**d**) plots obtained for UR12-P and UR12-PR samples immersed for different times in the test solution at OCP.

**Figure 12 materials-13-00738-f012:**
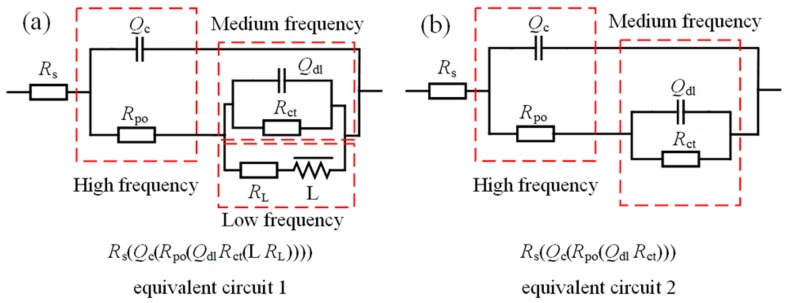
Two equivalent circuits used to fit the experimental electrochemical impedance spectroscopy (EIS) data of different samples. Circuit characterized by (**a**) two capacitive semicircles and an inductance and (**b**) two capacitive semicircles.

**Figure 13 materials-13-00738-f013:**
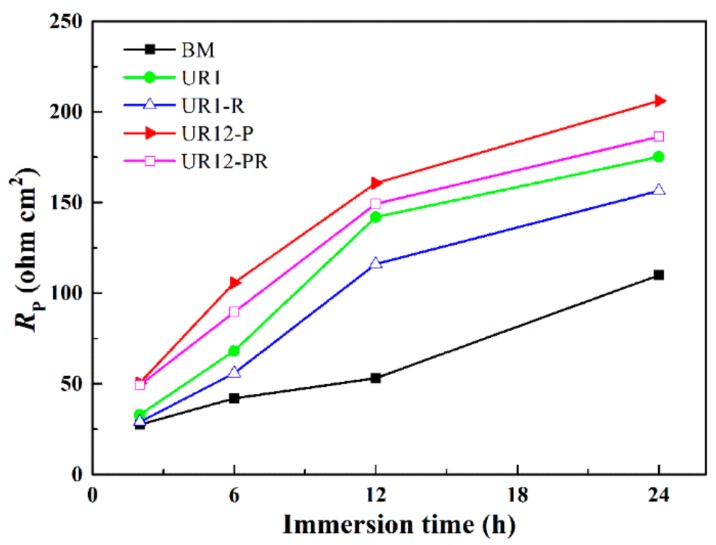
Dependence of the polarization resistance on time in the test solution at OCP.

**Table 1 materials-13-00738-t001:** Chemical composition (wt%) of 7B50-T7751 aluminum alloy.

Zn	Mg	Cu	Zr	Ti	Si	Fe	Al
6.2	2.1	2.0	0.08	0.05	0.04	0.03	Balance

**Table 2 materials-13-00738-t002:** Processing parameters utilized in ultrasonic surface rolling process (USRP).

Parameter Type	Value
CNC lathe rotating speed *V_1_* (rpm)	75
Tool feeding rate *V_2_* (mm/rev)	0.14
Ultrasonic vibration frequency (kHz)	28
Ultrasonic vibration amplitude (μm)	10
Static force (N)	500

**Table 3 materials-13-00738-t003:** Statistical parameters of corrosion depths for different samples.

Parameters	BM	UR1	UR1-R	UR12-P	UR12-PR
Mean (μm)	147.2	104.3	124.5	65.3	76.6
Standard (μm)	69.8	59.1	66.7	38.5	44.6
Minimum (μm)	50.8	21.6	43.1	8.5	15.6
1^st^ Quartile (μm)	93.6	56.9	65.2	32.1	45.0
Median (μm)	144.6	85.2	115.7	60.9	62.5
3^rd^ Quartile (μm)	178.4	145.1	160.8	92.2	113.3
Maximum (μm)	295.4	221.6	268.1	164.1	179.9

**Table 4 materials-13-00738-t004:** Electrochemical corrosion parameters of different samples.

Parameters	BM	UR1	UR1-R	UR12-P	UR12-PR
*E*_corr_ (mV_SCE_)	−737.8	−730.1	−730.1	−719.5	−721.0
*i*_corr_ (μA/cm^2^)	743.8	567.0	727.6	453.6	471.0
CR (mm/year)	8.099	6.175	7.924	4.940	5.129

Note: the unit of corrosion rate (CR) is converted from μm/year to mm/year due to the large calculated values.

**Table 5 materials-13-00738-t005:** Flat-band potential, *E_fb_*, and thickness, *δ*, of passive films formed on different samples.

Parameters	BM	UR1	UR1-R	UR12-P	UR12-PR
*E_fb_* (mV_SCE_)	−648.2	−653.9	−657.1	−628.3	−622.5
*δ* (nm)	2.32	2.99	2.66	3.60	3.42

**Table 6 materials-13-00738-t006:** Electrochemical parameters fitted from EIS data of samples immersed in the solution for 2 h.

Sample	*R*_s_ Ω cm^2^	*Y*_0_ of *Q*_c_ Ω^−1^ cm^−2^ s^n^	n of *Q*_c_	*R*_po_ Ω cm^2^	*Y*_0_ of *Q*_dl_ Ω^−1^ cm^−2^ s^n^	n of *Q*_dl_	*R*_ct_ Ω cm^2^	L H cm^2^	*R*_L_ Ω cm^2^
BM	3.49	6.78 × 10^−4^	0.66	25.15	3.74 × 10^−3^	0.84	27.79	18.85	2.45
UR1	2.82	3.15 × 10^−4^	0.71	25.71	4.61 × 10^−3^	0.71	44.25	27.92	8.48
UR1-R	5.76	2.88 × 10^−4^	0.74	25.48	4.42 × 10^−3^	0.77	29.13	28.08	3.90
UR12-P	3.01	2.45 × 10^−4^	0.72	26.11	5.12 × 10^−3^	0.55	56.35	38.38	42.67
UR12-PR	3.66	2.32 × 10^−4^	0.74	25.90	4.97 × 10^−3^	0.58	53.24	36.58	41.46

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
