# Peer review of "Effects of Ultrasonic Surface Rolling on the Localized Corrosion Behavior of 7B50-T7751 Aluminum Alloy"

_materials, 2020, doi:10.3390/ma13030738_

Round 1
Reviewer 1 Report
Introduction: Please provide information about the possible industrial application of this method. It seems that the method has purely academic nature and cannot be applied in the industry.
Table 1: Was chemical composition tested or it is presented according to the standard?
Chapter 2.1: Please describe what was the method for cutting out the rods out of plates. Please provide information about additional operations like polishing.
Chapter 2.1: Please provide information about how many specimens were tested by each method.
Fig. 2: The legend is no readable, please enlarge it.
Fig. 8: please mark standard deviation
Conclusions 2: please provide percentage information about improvement.
Author Response
Response to Reviewer 1 Comments
I am very grateful to your careful review for our manuscript. According to your advice, we have modified the relevant contents in the manuscript. The modifications are marked with different color for easy checking. The marking rules are as follows, the deleted contents are marked with a strikethrough, and your recommended changes are highlighted in yellow with red words. Your questions are answered one by one as shown below.
Point 1: Introduction: Please provide information about the possible industrial application of this method. It seems that the method has purely academic nature and cannot be applied in the industry.
Response 1: Although the current researches on USRP are mostly academic, they have laid the foundation for the application of USRP technology in aerospace structure. According to your comment, we have modified the last sentence of introduction as follows: “The present investigation was aimed at elucidating the application prospect of USRP in raising the localized corrosion resistance of aeronautical aluminum alloy structure (e.g., fuselage frame, skin and stringer of the wing).”
Point 2: Table 1: Was chemical composition tested or it is presented according to the standard?
Response 2: The chemical composition is presented according to the material report provided by Southwest Aluminum Group Co., Ltd. (China).
Point 3: Chapter 2.1: Please describe what was the method for cutting out the rods out of plates. Please provide information about additional operations like polishing.
Response 3: I am sorry for the unclear description of the method for cutting out the rods out of plates. The related content has been rewritten in the revision as follows: “In preparation for USRP, cuboid blanks were cut along the rolling direction of the plate via a line cutting machine. These blanks were then processed into rod specimens (length: 65 mm, diameter: 10 mm) by a lathe.” The chemical stripping method was thoroughly described in Ref [25] as follows: “surface material was removed step by step using a corrosive solution prepared by diluting 200 g of sodium hydroxide, 20 g of sodium sulfide, and 35 mL of triethanolamine into 1 L of reagent water. The material removal speed was about 30 μm/min, and black corrosion products were removed by immersing the samples in a 30% nitric acid solution for 5 s.” We have rewritten the content related with polishing as follows: “The removal method involved a combined treatment of chemical stripping (corrosive solution: 200 g/L sodium hydroxide + 20 g/L sodium sulfide + 35 mL/L triethanolamine) [25] and polishing (i.e., the stripped sample was rotated on a lathe at 100 rpm and polished with #3000 silicon carbide sandpaper).
Point 4: Please provide information about how many specimens were tested by each method.
Response 4: We have added the content at the end of Section 2.1 as follows: “Three parallel specimens (of the same type) were chosen for each measurement, and the average of the data was taken.”
Point 5: Fig. 2: The legend is no readable, please enlarge it.
Response 5: I am sorry for the unclear Figure 2. We have enlarge the legend and increased the resolution of this figure.
Point 6: Fig. 8: please mark standard deviation.
Response 6: We have marked the standard deviation in Figure 8.
Point 7: Conclusions 2: please provide percentage information about improvement.
Response 7: We have provided average pitting area percentage and mean corrosion depth for UR1 and UR12-P samples in Conclusions 2. These specific data are given to avoid duplication with the percentage data in Conclusions 3.
Reviewer 2 Report
The manuscript is well elaborated and written, the research has been thoroughly planned and made, the paper is interesting for all readers interested in localized corrosion of Al alloys, very important in particular for aircraft. Despite that, I have several comments.
Page 1 - Abstract: the authors write “The results revealed that this alloy is prone to pitting.”. The results of this research? It is generally known.
Page 2: it is “gage” instead of “gauge”
Page 2: the authors metion about applied stress relaxation method and demaged layer removal giving only references. Please say a little more, no pressing readers to look for the cited papers.
Page 2: There is no name of a company delivering the alloy (name, town, country).
Pages 2-4: There are no names of companies delivering the used equipment (name, town, country) for SEM, TEM, CLSM, MSF-3M, EDS, Parstat 2273, URSP device.
Page 3 (2.3): what is a reagent water? How the are of pitting and depth of pits were quantitatively calculated, with an eye, by a software?
Page 4: There are no names of companies delivering the softwares (name, town, country) such as PowerSuit, ZsimpWin.
Page 4: it is “and” instead of comma
Page 4: suggest change “shown” to “demonstrated” (twice “show” is too much in one sentence).
Page 4: The grain size before rolling has not been given. It is important to know to what extent the rolling decreased the grain size.
Page 5: it is “showed” instead of “show”
Page 7: it is “oxygen-atom content” instead of “atomic content of oxygen”
Even if the tests were repeated, no standard deviations were given for means for many results (roughness, stress, pitting percentages, OCP and so on) shown in the text.
Page 7: the authors write “This indicated that the nanocrystalline surface may promote
the formation of Al2O3·3H2O (rather than amorphous Al2O3 [1]) in the oxide layer during 24-h immersion testing.”, and after “this treatment transformed amorphous Al2O3 into a more stable oxide form in the surface film, thereby forming an effective anti-corrosion barrier.”. Please decide what is really formed, hydrated alumina or amorphous aluminum oxide? Besides, I do not think that the authors show or confirm the reliability of last sentence.
Page 12: after s 2-4 immersion instead of “after 2-h immersion}.
Pages 13-15: The applied process parameters (independent variables) are the ultrasonic surface rolling with two numbers of passages (one or twelfth) and the residual stress relaxation method (absent or present). The first variable decides on the roughness, grain size, likely surface morphology (the size of precipitates), the second – on the residual stress level and perhaps on microstructure. All these parameters affect the corrosion behavior as discussed. However, the discussion is to a great extent supported by another reports. Therefore, I have two comments:
The authors write “the Sa values of the samples decreased by 73.35% and 61.46%, respectively, relative to that of the untreated sample. The surface roughness and sensitivity to pitting attack of the treated samples are lower than those of the untreated samples; this may be associated with the harder penetration of water molecules, oxygen, and chloride ions into isolated regions of inhomogeneities”. What does it mean “harder penetration”? Why the authors do not take into account better (lower) electrical homogeneity of surface for polished samples? If inhomogeneities are isolated, by what? How such penetration looks like? The authors write “The occurrence of an inductive circular arc in the impedance response of the UR1-R sample further confirmed that this stress played a significant role in inhibiting pit propagation”. Again, please propose the model demonstrating how the compressive stress may inhibit the crack propagation.The manuscript is prepared in almost excellent English. I would only like to suggest authors to take a care about a use of different tenses in the same sentence (e.g. pages 4, 5, 7).
Author Response
Response to Reviewer 2 Comments
I am very grateful to your careful review for our manuscript. According to your advice, we have modified the relevant contents in the manuscript. The modifications are marked with different color for easy checking. The marking rules are as follows, the deleted contents are marked with a strikethrough, and your recommended changes are highlighted in yellow. Your questions are answered one by one as shown below.
Point 1: Page 1 Abstract: the authors write “The results revealed that this alloy is prone to pitting.”. The results of this research? It is generally known.
Response 1: 7B50-T7751 aluminum alloy supplied by Southwest Aluminum Group Co., Ltd. (Chongqing, China) is a newly developed 7000 series aluminum alloy. It has undergone the retrogression and re-ageing treatment and is theoretically insensitive to intergranular corrosion. Since detailed corrosion data of this alloy in a sodium chloride + hydrogen peroxide solution are rarely reported in other literatures, we provide its pitting corrosion data. Additionally, for the purpose of comparison with USRP-treated samples, we stated that the alloy is prone to pitting in this corrosion solution, thereby highlighting the central idea of this article, i.e., USRP treatment can improve the local corrosion resistance of 7B50 aluminum alloy.
Point 2: Page 2: it is “gage” instead of “gauge”.
Response 2: According to your advice, we have replaced gage with gauge.
Point 3: Page 2: the authors mention about applied stress relaxation method and damaged layer removal giving only references. Please say a little more, no pressing readers to look for the cited papers.
Response 3: According to your suggestion, we have supplemented more information on stress relaxation process and removal method. Besides, we have modified the statement describing the residual stress relaxation process as follows: “The effect of residual stress on the corrosion behavior of the USRP-treated alloy was determined by performing a residual stress relaxation process on circular cross-sectional fatigue specimens (gauge length: 35 mm, diameter: 10 mm).”
Point 4: Page 2: There is no name of a company delivering the alloy (name, town, country).
Response 4: We have supplemented information on the manufacturer of the delivered alloy as follows: “In this study, the commercial 7B50-T7751 aluminum alloy (Southwest Aluminum Group Co., Ltd., Chongqing, China) with a retrogression and re-ageing treatment was employed.”
Point 5: Pages 2-4: There are no names of companies delivering the used equipment (name, town, country) for SEM, TEM, CLSM, MSF-3M, EDS, Parstat 2273, URSP device.
Response 5: We have supplemented information on the manufacturers of these instruments as follows: the USRP device (HK30G, Shandong Huawin Electrical & Mechanical Technology Co., Ltd, Shandong, China), SEM (Vega II, Tescan, Kohoutovice, Czech Republic), CLSM (LSM 700, Carl Zeiss AG, Oberkochen, Germany), TEM (Tecnai G2 F20, FEI, Oregon, USA), MSF-3M (Rigaku, Tokyo, Japan), EDS (INCA Energy 350 EDX analyzer, Oxford Instruments, Oxfordshire, UK), and Parstat 2273 (AMETEK, Inc., Pennsylvania, USA).
Point 6: Page 3 (2.3): what is a reagent water? How the area of pitting and depth of pits were quantitatively calculated, with an eye, by a software?
Response 6: Reagent water is the most widely used analytical solvent in a clinical laboratory. In fact, we use distilled water instead of reagent water to prepare test solution, so the term "reagent water" is misused in this article. We have replaced it with “distilled water”. We utilized Image-Pro Plus software (Media Cybernetics, Inc., Maryland, USA) to quantitatively calculate the area of pitting and measure the depth of pits. Related content has been added to Section 2.3.
Point 7: There are no names of companies delivering the software (name, town, country) such as PowerSuit, ZsimpWin.
Response 7: We have supplemented information on the software as follows: PowerSuite software (Princeton Applied Research, New Jersey, USA) and ZsimpWin software (EChem Software, Michigan, USA).
Point 8: Page 4: it is “and” instead of comma.
Response 8: We have made corresponding modification as follows: “Where, f = 1000 Hz, Zim is the imaginary part of the impedance.” However, I'm not quite sure if this is what you expected me to modify.
Point 9: Page 4: suggest change “shown” to “demonstrated” (twice “show” is too much in one sentence).
Response 9: According to your advice, we have replaced “shown” with “demonstrated”.
Point 10: Page 4: The grain size before rolling has not been given. It is important to know to what extent the rolling decreased the grain size.
Response 10: A corroding to your suggestion, we have supplemented the grain size before USRP treatment in Section 2.1 as follows: “The mean grain size ranges were 500~2000 μm, 50~120 μm and 5~25 μm, in the longitudinal, transverse and thickness directions, respectively.”
Point 11: Page 5: it is “showed” instead of “show”
Response 11: According to your advice, we have located only one sentence containing “show” on page 5 as follows: “Insets show the corresponding SAED patterns.” The tense of the figure caption is usually the present simple tense, so it is better to use “show”.
Point 12: Page 7: it is “oxygen-atom content” instead of “atomic content of oxygen”.
Response 12: According to your suggestion, we have replaced “oxygen-atom content” with “atomic content of oxygen”.
Point 13: Even if the tests were repeated, no standard deviations were given for means for many results (roughness, stress, pitting percentages, OCP and so on) shown in the text.
Response 13: According to your advice, we have supplemented the standard deviations of roughness, stress, pitting area percentages and corrosion depth.
Point 14: Page 7: the authors write “This indicated that the nanocrystalline surface may promote the formation of Al2O3·3H2O (rather than amorphous Al2O3 [1]) in the oxide layer during 24-h immersion testing.”, and after “this treatment transformed amorphous Al2O3 into a more stable oxide form in the surface film, thereby forming an effective anti-corrosion barrier.”. Please decide what is really formed, hydrated alumina or amorphous aluminum oxide? Besides, I do not think that the authors show or confirm the reliability of last sentence.
Response 14: Existing literatures show that the oxide layer of aluminum alloy is divided into inner layer and external layer. The inner layer is generally composed of amorphous Al2O3. Through EDS analysis, it can be seen that the atomic content of oxygen of BM and UR1 samples is basically the same, that is, the composition of the inner layers of the two samples have not changed. However, the atomic content of oxygen in the UR12-P sample increase significantly. We speculate that the inner layer of the UR12-P sample may be Al2O3·3H2O. Furthermore, “this treatment transformed amorphous Al2O3 into a more stable oxide form in the surface film, thereby forming an effective anti-corrosion barrier.” is a rewrite of the sentence in Ref [11]. Since our expression is not clear and may cause you misunderstanding, we have modified this sentence as follows: “Similarly, Trdan [11] evaluated the corrosion resistance of laser shock peened AA6082-T651 aluminum alloy after 24 h of exposure to 0.6M NaCl. He found that this treatment transformed amorphous Al2O3 into a more stable oxide form in the surface film, thereby forming an effective anti-corrosion barrier.”
Point 15: Page 12: “after s 2-4 immersion” instead of “after 2-h immersion”.
Response 15: Thanks for your careful review. We have corrected this low-level error.
Point 16: Pages 13-15: The applied process parameters (independent variables) are the ultrasonic surface rolling with two numbers of passages (one or twelfth) and the residual stress relaxation method (absent or present). The first variable decides on the roughness, grain size, likely surface morphology (the size of precipitates), the second – on the residual stress level and perhaps on microstructure. All these parameters affect the corrosion behavior as discussed. However, the discussion is to a great extent supported by another reports. Therefore, I have two comments:
The authors write “the Sa values of the samples decreased by 73.35% and 61.46%, respectively, relative to that of the untreated sample. The surface roughness and sensitivity to pitting attack of the treated samples are lower than those of the untreated samples; this may be associated with the harder penetration of water molecules, oxygen, and chloride ions into isolated regions of inhomogeneities”. What does it mean “harder penetration”? Why the authors do not take into account better (lower) electrical homogeneity of surface for polished samples? If inhomogeneities are isolated, by what? How such penetration looks like? The authors write “The occurrence of an inductive circular arc in the impedance response of the UR1-R sample further confirmed that this stress played a significant role in inhibiting pit propagation”. Again, please propose the model demonstrating how the compressive stress may inhibit the crack propagation.
Response 16: Firstly, the purpose of this article is to study the corrosion resistance of 7B50 aluminum alloy with a new surface processing technology. Not only the new technology itself but also the research results of similar technologies should be compared and analyzed in order to comprehensively understand the characteristic of the new technology. This is the general research method. In this paper, we use this method to discuss the pitting behavior of 7B50 alloy subjected to USRP treatment. It is understandable to support my work with published opinions.
Secondly, the “harder penetration” means difficult to penetrate, i.e., water molecules, oxygen, and chloride ions are difficult to penetrate into isolated regions of inhomogeneities in the smooth surface comparing with that in rough surface. Based on published opinions, we explain the possible results of the experimental measurement data in this study. This article focuses on the effects of compressive residual stress, grain refinement and surface-layer microstructural homogenization on the corrosion resistance of USRP-treated 7B50 aluminum alloy. According to the test results in this article, we can see that these three factors are important factors. Although roughness is also an influencing factor, this article does not conduct an in-depth study of roughness because it has a small degree of influence. The isolated regions of inhomogeneities refer to intermetallics, passive film defects, pores, etc. We have added this content in the article. Unfortunately, we do not yet know of any effective technology to isolate these regions.
Finally, thank you for your valuable suggestion. Obviously, it is beneficial to control pitting corrosion to propose a model in which compressive residual stress may inhibit crack propagation. At present, our research is limited to discovering the effect of compressive residual stress. As for the establishment of the specific model, further analysis is needed, which will be our future research direction. Besides, the term “confirmed” may be misleading to readers, and we have replaced it with “showed”.
Point 17: The manuscript is prepared in almost excellent English. I would only like to suggest authors to take a care about a use of different tenses in the same sentence (e.g. pages 4, 5, 7).
Response 17: Thank you for your suggestion, we are overly pursuing the consistency of the tense of sentence, and we will fully show the flexibility of expression in future writing.
Reviewer 3 Report
Perhaps, a study of the concentration in a model solution of elements leached from an alloy during corrosion (the latter are missing in the manuscript) could improve complexity of the work. It is also interesting whether the possibility of corrosion of the entire sample not just in the form of separate pitting holes was considered - was the change in the size of the sample determined after testing?
Anyway, the manuscript is acceptable for publishing in the original form, after slight revisions are done. My comments are available in the attached document “materials-711891-peer-review-v1-2.pdf”.

Author Response
Response to Reviewer 3 Comments
I am very grateful to your careful review for our manuscript. According to your advice, we have modified the relevant contents in the manuscript. The modifications are marked with different color for easy checking. The marking rules are as follows, the deleted contents are marked with a strikethrough, and your recommended changes are highlighted in bright green. Your questions are answered one by one as shown below.
Point 1: Perhaps, a study of the concentration in a model solution of elements leached from an alloy during corrosion (the latter are missing in the manuscript) could improve complexity of the work. It is also interesting whether the possibility of corrosion of the entire sample not just in the form of separate pitting holes was considered - was the change in the size of the sample determined after testing?
Response 1: This article lacks the research on the dissolution model of elements leached from an alloy during corrosion. It is a very meaningful study. The reviewer's suggestion will provide important guidance for our future research, and we are grateful for this. We considered the possibility of corrosion of the entire sample and determined the size of samples after a 24-hour test. The dimensional changes of different samples were not obvious.
Point 2: Who produces the plate? Need to specify the manufacturer.
Response 2: According to your suggestion, information on 7B50 aluminum alloy manufacturer has been added in the article. The plate was provided by Southwest Aluminum Group Co., Ltd. (Chongqing, China).
Point 3: Why this treatment is only for the 12-fold method. For comparison, it was not necessary to do it for a one-time too?
Response 3: The removal treatment was aimed at eliminating the micro-damage caused by the 12-pass URSP treatment. After this treatment the 12-pass USRP-treated samples still retain a sufficiently thick deformation strengthening layer. However, for one-pass USRP-treated samples, the removal treatment will weaken the favorable factors introduced by USRP. Considering this, the treatment is only for the 12-fold method.
Point 4: Hereinafter: manufacturer - company, country, city.
Response 4: We have added the manufacturer information for each instrument in Section 2.
Reviewer 4 Report
Generally this is a paper of good scientific quality that should be published. However there are a few minor points that would strengthen an already good report.
Firstly - in the keywords should USRP be replaced with ultrasonic surface rolling process?
The introduction is well written and comprehensive.
It would aid the readers to add a sentence to overview the stress relaxation process as well as providing the reference.
What does BM stand for?
Following the cutting to 15 mm, how were the sample surfaces treated?
How were the elemental compositions obtained? And what are the confidence intervals in these values?
Is the ultrasonic surface rolling device built in house or is it a commercial machine? Please add details.
What is the estimated spot size of the XRD?
What EDS parameters were used? Exposure time, voltage, current, spot size
A sentence after the 'The influence of USRP on the surface...' to overview what was discussed would be useful.
The font used on the subplots in figure 2 (a,b,c,d,f) is too small to read
Was the XRD performed at one spot on each sample? Are the results repeatable at different locations?
The font on the EDS spectra in Figure 5 are too small to read
The font size on Figure 6a,b,g and h is too small to read
Please avoid starting paragraphs with Figure X, Figure Y...
The paper contains many figures - are the authors sure all are needed?
A paragraph on the wider implications of the research is missing from the discussion.
Author Response
Response to Reviewer 4 Comments
I am very grateful to your careful review for our manuscript. According to your advice, we have modified the relevant contents in the manuscript. The modifications are marked with different color for easy checking. The marking rules are as follows, the deleted contents are marked with a strikethrough, and your recommended changes are highlighted in cyan. Your questions are answered one by one as shown below.
Point 1: Firstly - in the keywords should USRP be replaced with ultrasonic surface rolling process?
Response 1: According to your suggestion, we have replaced USRP with ultrasonic surface rolling process.
Point 2: It would aid the readers to add a sentence to overview the stress relaxation process as well as providing the reference.
Response 2: Thank you for your valuable suggestion, we have supplemented the detailed stress relaxation process in the revision as follows: “This process was conducted using 0.1 Hz sine wave with stress ratio =−1 under a maximum stress of 350 MPa and was run for 10 cycles on a fatigue testing machine (SDS-100, Changchun Research Institute for Mechanical Science Co., Ltd, Changchun, China).”
Point 3: What does BM stand for?
Response 3: In Section 2.1, we wrote “The samples without USRP treatment were referred to as BM.” We have highlighted this sentence for your review.
Point 4: Following the cutting to 15 mm, how were the sample surfaces treated?
Response 4: Before cutting to 15 mm, all samples have subjected to USRP treatment except for BM samples. After cutting, these samples maintained their respective surface conditions.
Point 5: How were the elemental compositions obtained? And what are the confidence intervals in these values?
Response 5: The chemical composition is presented according to the material report provided by Southwest Aluminum Group Co., Ltd. (China).
Point 6: Is the ultrasonic surface rolling device built in house or is it a commercial machine? Please add details.
Response 6: The ultrasonic surface rolling device is a commercial machine. We have supplemented its details as follows: “A schematic of the USRP device (HK30G, Shandong Huawin Electrical & Mechanical Technology Co., Ltd, Shandong, China) is shown in Figure 1,”
Point 7: What is the estimated spot size of the XRD?
Response 7: We are so sorry, we only input the parameters of tube voltage (40 kV), tube current (40 mA), scanning angles (30–85°), scanning rate (8°/min), and interval step (0.02°/s) in the software during XRD measurement. By consulting the data, it can be estimated that the spot size is not less than 20 μm.
Point 8: What EDS parameters were used? Exposure time, voltage, current, spot size.
Response 8: The detailed EDS parameters were: exposure time (30s), acceleration voltage (20 kV), emission current (10 μA). Since the sampling area is manually selected, the spot size is approximately ~15 μm.
Point 9: A sentence after the 'The influence of USRP on the surface...' to overview what was discussed would be useful.
Response 9: According to your advice, we have supplemented the content after the 'The influence of USRP on the surface...' as follows: “For example, the surface roughness and damage increased with the number of USRP treatment.”
Point 10: The font used on the subplots in figure 2 (a,b,c,d,f) is too small to read.
Response 10: We have enlarge the subplots in Figure 2 and increased the resolution of this figure.
Point 11: Was the XRD performed at one spot on each sample? Are the results repeatable at different locations?
Response 11: In fact, the XRD performed at two spots on each sample. The XRD peak shape and position are repeatable.
Point 12: The font on the EDS spectra in Figure 5 are too small to read.
Response 12: We have enlarge the font on the EDS spectra in Figure 5 and increased the resolution of this figure.
Point 13: The font size on Figure 6a,b,g and h is too small to read.
Response 13: We have enlarge the font on Figure 6 and increased the resolution of this figure.
Point 14: Please avoid starting paragraphs with Figure X, Figure Y...
Response 14: Thank you for your valuable advice, we will avoid writing this way in the future paper.
Point 15: The paper contains many figures - are the authors sure all are needed?
Response 15: This article wants to show as much as possible the effect of USRP treatment on the corrosion resistance of 7B50 aluminum alloy, so many graphs and data are given. We think they are all necessary, and we believe some readers will get inspiration from these graphs and data.
Point 16: A paragraph on the wider implications of the research is missing from the discussion.
Response 16: Due to space limitations, the discussion section only discusses the effect of USRP treatment on pitting initiation and propagation. We will discuss the wider implications of the research in the next paper.